# Enhanced Molybdenum Recovery Achieved by a Complex of Porous Material-Immobilized Surface-Engineered Yeast in Development of a Sustainable Biosorption Technology

**DOI:** 10.3390/microorganisms13051034

**Published:** 2025-04-30

**Authors:** Thiti Jittayasotorn, Kentaro Kojima, Audrey Stephanie, Kaho Nakamura, Hernando P. Bacosa, Kengo Kubota, Masanobu Kamitakahara, Chihiro Inoue, Mei-Fang Chien

**Affiliations:** 1Department of Environmental Studies for Advanced Society, Graduate School of Environmental Studies, Tohoku University, 6-6-20 Aoba, Aramaki, Aoba-ku, Sendai 980-8579, Japan; thiti.jittayasotorn.s5@dc.tohoku.ac.jp (T.J.); masanobu.kamitakahara.a6@tohoku.ac.jp (M.K.); chihiro.inoue.b1@tohoku.ac.jp (C.I.); 2Department of Environmental Science, School of Interdisciplinary Studies, Mindanao State University-Iligan Institute of Technology, Andres Bonifacio Avenue, Iligan 9200, Philippines; hernando.bacosa@g.msuiit.edu.ph; 3Department of Frontier Sciences for Advanced Environment, Graduate of Environmental Studies, and Department of Civil and Environmental Engineering, Graduate School of Engineering, Tohoku University, 6-6-06 Aoba, Aramaki, Aoba-ku, Sendai 980-8579, Japan; kengo.kubota.a7@tohoku.ac.jp

**Keywords:** bio-recovery, molybdate, rare metal recovery, immobilization, cell-surface display, *Saccharomyces cerevisiae*

## Abstract

Molybdenum (Mo) is a critical industrial metal valued for its corrosion resistance and strength-enhancing properties. However, increasing demand necessitates more efficient and sustainable recovery methods. Bio-recovery of Mo by biosorption is a promising resolution, especially by the use of surface-engineered microbes that express metal binding proteins on its cell surface. This study investigates the potential of *Saccharomyces cerevisiae* strain ScBp6, which displays a molybdate-binding protein (ModE) on its cell surface, immobilized on porous materials. Our findings reveal that polyurethane sponges (PS) significantly outperform ceramic materials in yeast immobilization, entrapping 1.76 × 10^7^ cells per sponge compared to 1.70 × 10^6^ cells per ceramic cube. Furthermore, the yeast–PS complex demonstrated superior Mo adsorption, reaching 2.16 pg Mo per yeast cell under 10 ppm Mo conditions, comparable to free yeast cells (1.96 pg Mo per yeast cell). These results establish PS as an effective and scalable platform for Mo recovery, offering high biosorption efficiency, reusability, and potential for industrial wastewater treatment applications.

## 1. Introduction

Molybdenum (Mo) is a vital trace element widely utilized in fertilizers, catalysts, metal alloys, and anti-corrosive agents. Its essential role in stainless steel manufacturing underscores its industrial significance [1,2,3,4]. Despite its recyclability, Mo recovery remains inefficient; in Japan, only 2% of Mo is recovered through recycling, with most Mo used in steel products being reclaimed from scrap metal [5]. However, a substantial portion of Mo, ranging from 10 µg/L to 10 mg/L, is found in industrial wastewater, presenting an untapped opportunity for resource recovery [6]. According to Henckens et al., only about 20% of end-of-life Mo is recycled globally, while the remaining 80% is dissipated, down-cycled into lower-grade steels, or lost during production [7]. Additionally, the International Molybdenum Association estimates that approximately 25,300 tons of Mo, or around 10% of global production, ends up in wastewater at low concentrations each year [6]. This contamination underscores the urgent need for effective methods to remove and recover Mo from polluted water sources [3].

Mo-contaminated wastewater from industrial activities could serve as a potential source for recovery [8]. However, practical applications for Mo recovery are challenging due to its low concentration in wastewater [9]. Several studies regarding metal recovery operations have already been developed and utilized in various industries. Physical and chemical methods, such as chemical precipitation, reverse osmosis, and ion exchange, are still facing challenges related to environmental impact and operation costs [8,9,10,11]. Given the limitations of existing recovery methods, innovative approaches are required to enhance the efficiency of Mo recovery. One promising avenue that does not require expensive cost and is environmentally friendly is through biological methods, such as biosorption, as Mo is one of the important components in biology [12]. Several living organisms require Mo as a co-factor for their enzymes. For example, archaebacteria have the ability to uptake and transport various molecules in the environment into their cell through the ABC-transport operon [13].

Among promising biosorption tools, cell-surface engineering, which enables the modification of cell membranes with functional biomolecules, such as peptides or metal-binding proteins, through genetic or chemical means to impart programmable interactions and enhance environmental responsiveness [14], has enabled the development of microorganisms that express specific metal-binding proteins on their outer membranes. For instance, *ModE*, a molybdate-responsive transcriptional regulator found in bacteria, binds molybdate ions (MoO_4_^2−^) with high specificity [15,16]. Hui et al. showed that *E. coli* displaying the Pb^2+^-binding domain of PbrR adsorbed nearly twice as much lead as cells displaying the full-length protein. The engineered cells also maintained high selectivity in the presence of Zn^2+^ and Cd^2+^ [17]. Similarly, Thai et al. used a surface display library to isolate peptides that strongly bind to Cu_2_O and ZnO. The selected sequences showed clear patterns of metal-specific binding, linked to the position of basic residues, like arginine and lysine [18]. Smith et al. tried to create a fusion protein by inserting the phage into *geneIII*, which resulted in a fusion phage that displayed the foreign amino acid in an immunologically accessible form [19]. Due to this finding, several scientists later developed a cell-surface displayed on the microorganism. Yeasts are widely used in biotechnology due to their fast growth, environmental tolerance, and ease of genetic modification [20]. Among them, *Saccharomyces cerevisiae* provides an advantageous platform for surface engineering due to its genetic tractability and robustness under diverse environmental conditions, facilitating practical biosorption applications. Besides, *S. cerevisiae* does not metabolize Mo, so the effect of engineered yeast on Mo biosorption is clear. Through this cell-surface displayed technology, our previous study successfully constructed a cell-surface displayed yeast that expressed the C-terminal part of ModE protein, and further modification for a better expression has been performed [11,20,21]. However, the practical use of the cell-surface engineered yeast at an industrial scale is still a challenge.

Similar challenges have been reported in bacterial systems, where free-cell applications suffer from low biomass retention and stress susceptibility, leading to reduced performance. Immobilization strategies have been proposed as a promising solution to enhance stability and metal removal efficiency in such systems [22]. Wong et al. studied the immobilization of a poly-acramide gel of *Pseudomonas putida* in copper (Cu) recovery, which resulted in more than 90% of Cu(II) adsorbed on immobilized cells being recovered by eluting with 0.1 M HCl [23]. More recent advances in immobilized biosorbents have reinforced this strategy’s industrial relevance. Sun et al. achieved over 97% removal of La, Y, and Sm by immobilizing *Galdieria sulphuraria* in calcium alginate beads, while simultaneously enhancing ammonium removal from acidic wastewater [24]. Effective immobilization depends largely on the interaction between the microbial surface and the material, making high-surface-area structures essential for stable and efficient cell retention. Immobilization using porous materials is particularly promising. These supports offer large surface areas, facilitate cell attachment, and enable nutrient diffusion.

These studies highlight the potential of immobilized biosorbents to perform effectively under complex wastewater conditions. In particular, systems such as the Downflow Hanging Sponge (DHS) reactor benefit from high-surface-area supports, like polyurethane sponges (PS), which are well-suited for continuous flow operation. Besides, to our knowledge, currently there are no studies that have explored the integration of surface-engineered yeast with porous material immobilization for rare metal biosorption, particularly in the context of continuous treatment systems.

In this study, we investigated the use of ModE surface engineered yeast cells called *Saccharomyces cerevisiae* strain ScBp6, immobilized on three types of porous materials: porous ceramic plates with different pore sizes and PS, a support widely used in biofilm reactors, which has been applied for microbial retention in continuous wastewater treatment processes as the DHS system [25]. The ScBp6 strain was engineered to display the C-terminal domain of the bacterial ModE protein on its cell surface. ModE is a molybdate-responsive transcriptional regulator that binds MoO_4_^2−^ with high specificity [11]. By leveraging this binding domain, ScBp6 is capable of selectively adsorbing Mo from solution without requiring intracellular metabolism, enabling an efficient biosorption mechanism. In addition, porous materials play a crucial role in immobilization by providing physical support and am high surface area that facilitate cell attachment and proliferation. In particular, materials with interconnected pore networks can enhance microbial retention by minimizing washout, while also improving nutrient diffusion. Therefore, selecting a material with appropriate porosity and structural integrity is essential for stable and effective biosorption systems. This method leverages the biological affinity of engineered yeast for Mo, combined with the high surface area and adsorption capacity of porous materials, to facilitate the effective recovery of Mo from contaminated water sources. This approach not only addresses the environmental impact of Mo contamination but also contributes to the sustainable management of this critical resource.

## 2. Materials and Methods

### 2.1. Yeast Strain and Cultivation

In this study *Saccharomyces cerevisiae* strain ScBp6 was utilized in all experiments. Yeast strain ScBp6 was constructed in our laboratory as described in [11]. The culture was carried out in SD medium without leucine (SD(-leu)). The yeast strain ScBp6 was stored frozen in a 25% glycerol solution at −80 °C until use. For cultivation, the frozen yeast was added at a concentration of 1% (*v*/*v*) to 15 mL of SD(-leu) medium in a centrifuge tube, and pre-cultured at 30 °C and 150 rpm for 24 h. Yeast cell counts were determined using the Microbial Viability Assay Kit-WST (M439, DO-JINDO, Kumamoto, Japan). Absorbance was measured at 450 nm using a Varioskan™ LUX multimode microplate reader (Thermo Fisher Scientific Inc. Waltham, MA, USA). Then, a calibration curve was generated using serial dilutions and direct cell counting under a microscope to ensure accurate quantification.

### 2.2. Experimental Design

This study was conducted in two main phases to evaluate the performance of surface-engineered yeast ScBp6 for Mo biosorption. In the first phase, three types of ceramic materials were tested for yeast immobilization: P40, P45, and a dense Plate. Yeast adhesion, retention, and proliferation were assessed for each ceramic type. These ceramic materials offered controlled pore size variation, but initial results showed relatively low yeast retention across all ceramic types. Based on this observation and informed by previous studies demonstrating its effectiveness in biofilm reactors [25], PS was selected for further evaluation.

The PS was evaluated for its ability to retain immobilized ScBp6 yeast and support growth. In the second phase, Mo adsorption experiments were conducted using immobilized yeast on PS and free ScBp6 cells in parallel. ICP–MS (ICP–MS Nexion, Perkin Elmer, Waltham, MA, USA) was used to determine Mo concentration, and adsorption kinetics were monitored over time. Scanning electron microscopy (SEM; SU8000, Hitachi High-Tech Co., Tokyo, Japan) was employed to analyze material surface structure, and all experiments were performed in triplicate with appropriate statistical analysis.

### 2.3. Immobilization Materials and Yeast Immobilization

#### 2.3.1. Ceramic Materials

Two types of porous ceramics and dense body ceramics were prepared to understand how different shapes of fixative materials affect the adhesion of yeast. Commercially available ceramics, with varying pore size diameter, were used in this study, labeled as P40 (Ø; 50–100 µm), P45 (Ø; 300–1000 µm) and Plate (dense) (AS ONE Corporation, Osaka, Japan). In all experiments involving ceramic porous material, the materials were wrapped in aluminum foil, sealed in plastic bags, and autoclaved at 121 °C for 20 min. The ceramics are plate-like, measuring 25 mm on each side and 5 mm in thickness.

In the yeast immobilization experiments, three ceramics of each type were placed in a plastic container. Each vessel, including a control without ceramics, received 30 mL of yeast suspension adjusted to an optical density (OD) at 600 nm of 1.0. Immobilization was carried out for 1 h with shaking at 70 rpm. The samples were then washed twice with 10 mL of Tris–HCl buffer while shaking and then transferred to a new petri dish. Ten mL of a test solution mixed with Tris–HCl buffer and assay kit in a ratio of 190:10 were added incrementally. After 30 min, 200 µL of the mixture was collected at a time, and absorbance was measured using a microplate reader. Yeast cells were collected from the washing solution and immobilization solutions, and their absorbance were measured. The number of immobilized yeast cells was estimated based on a standard calibration curve correlating absorbance at 450 nm with known yeast concentrations. This curve was generated from serial dilutions and direct microscopic cell counts. Yeast adhesion was evaluated after 1 h of immobilization, while proliferation was assessed by measuring the increase in cell numbers after 24-h incubation in SD medium.

#### 2.3.2. Polyurethane Sponges

Polyurethane sponges (PS) were obtained from Kubota et al., 2024 [25]. This PS material has been proven to immobilize microorganisms in the sewage treatment systems known as Downflow Hanging sponge (DHS) [25], a biofilm reactor widely applied for wastewater treatment. Then, the pore size of PS was measured in this study. In this experiment, 65 mL of the yeast solution (OD_600nm_ = 1.0) was placed in a food container and a PS was submerged. The yeast was immobilized by shaking the container at 30 °C and 100 rpm for 1 h. After that, the PS was then removed from the solution, and the yeast within the PS was squeezed out into a separate container. The squeezed yeast was replaced with PBS solution. The sponges were repeatedly squeezed into PBS five times, collecting all detached cells and ensuring that all yeast was removed. A total of 190 mL of the cells in PBS solution obtained from the above steps was mixed with 10 mL of the Assay Kit-WST reagent, then added to each well of a 96-well microplate. After mixing and incubating at 30 °C for 30 min, the absorbance at 450 nm was measured. As described above, cell numbers were quantified using a standard calibration curve based on WST absorbance and direct microscopic counts. Immobilization and proliferation were assessed at 1 and 24 h, respectively. In addition, PS retention capability was evaluated by measuring yeast efflux into the PBS after 24-h incubation under shaking at 30 °C.

#### 2.3.3. Surface Characterization of Porous Material

To evaluate the structural characteristics of the porous materials used in yeast immobilization, scanning electron microscopy (SEM) analysis was conducted. The surfaces of the ceramic and polyurethane sponge samples were imaged using SEM at an accelerating voltage of 5.0 kV. This imaging allowed for a detailed examination of the pore morphology, surface roughness, and potential binding sites that contribute to yeast adhesion and immobilization.

### 2.4. Mo Adsorption by Immobilized Yeast

The immobilized yeast on PS (as described in steps 2/3) and an equivalent amount of free yeast (1.76 × 10^8^ cells) were subjected to Mo-solution with Mo concentrations adjusted to 10 ppm, using ultrapure water for the Mo adsorption experiments. The concentration of 10 ppm Mo was chosen to simulate upper-range industrial wastewater concentrations (typically reported between 10 µg/L and 10 mg/L), as previously documented [5,7]. The Mo-solution volume was immobilized at 90 mL, and the experiments were conducted at 30 °C and 100 rpm. The experiment was performed in triplicate. Immediately after the start (0 h) and at 1, 2, 6, and 24 h, 100 µL of Mo-solution was removed from the vessel. ScBp6-free cells were centrifuged at 6000 rpm for 5 min, and then the precipitated yeast and Mo-solution were collected. Then, collected Mo-solution samples were diluted to provide Mo concentration between 3 and 20 ppb, and the ICP–MS solution for Mo adsorption measurement was prepared. The mixture of 500 µL of 60% nitric acid and 10 µL of indium at 10 ppm was prepared as an internal standard. Then, this was adjusted to a final volume of 10 mL with MilliQ water in a volumetric flask. For free yeast, 3 mL of 60% nitric acid was added to the centrifuged ScBp6-free cell pellet, then pyrolyzed at 130 °C for 1 h. The prepared ICP–MS solution was filtered through a 0.45 µm filter. The Mo concentration was then measured by ICP–MS.

### 2.5. Statistical Analysis

All experiments were conducted in triplicate (*n* = 3). Data are presented as mean ± standard deviation (SD). Statistical significance was evaluated using one-way analysis of variance (ANOVA), with *p* < 0.05 considered significant. Graphs were generated with error bars representing SD to indicate data variability.

## 3. Results

### 3.1. Yeast Immobilization

#### 3.1.1. Yeast Immobilization on Porous Ceramics

To enhance the available surface area for yeast immobilization, porous ceramic materials were employed in this study. Figure 1 presents the macroscopic and microscopic characteristics of the ceramic materials investigated: P40, P45, and a non-porous plate. Figure 1A shows the gross morphology of the ceramic materials used for yeast immobilization. While pore structures are not apparent in the digital photographs, these images are provided to illustrate the size, shape, and general appearance of the ceramic formats used in the experiments. The P40 and P45 exhibit visibly porous structures, while the plate shows no observable porosity. Figure 1B displays SEM images of the ceramics at 50× magnification, revealing distinct surface morphologies. The P40 ceramic (left) exhibits a highly interconnected and uniform pore network with pore sizes around 100 µm, which could facilitate stable yeast attachment by providing consistent anchoring sites. In contrasts, the P45 ceramic (right) features larger, more irregularly shaped pores ranging from 300 to 1000 µm, forming a deeper and more open structure. This increased pore size may allow for higher yeast cell immobilization but could also lead to weaker retention, as observed in the washing results. The non-porous plate, lacking any internal structure for cell attachment, predictably showed no yeast immobilization within its material.

These morphological characteristics directly correlate with the yeast immobilization efficiency observed in Figure 2. The smaller, more uniform pores of P40 enabled stronger yeast adhesion, leading to greater retention during washing. Meanwhile, the larger and deeper pore structures of P45 allowed for a higher overall immobilization density but at the cost of increased yeast detachment. The absence of porosity in the plate completely inhibited yeast entrapment, confirming the necessity of a porous matrix for effective immobilization.

Based on the results shown in Figure 2, the total number of yeast cells in the solution is the sum of the immobilized yeast cells, yeast cells in the washing solution, and yeast cells in the immobilization solution. Approximately 0.15 × 10^7^ yeast cells from the total population were immobilized in ceramics with P40, while ceramics with P45 immobilized approximately 0.17 × 10^7^ yeast cells. In contrast, the non-porous plate showed no yeast immobilization within its structure.

When analyzing the number of yeast cells in the washing solution, it is evident that ceramics with P45 retain a significant amount of yeast during the initial immobilization stage. However, these results also reveal that a considerable proportion of yeast initially adhering to the P45 ceramic is subsequently washed away during the washing process. Conversely, P40 ceramics demonstrate a higher initial adhesion capacity, with fewer yeast cells being washed away, indicating more stable attachment.

Taking into account the total immobilized yeast cell count, including those washed away, the ceramics with P45 exhibit a slightly higher overall retention of yeast cells (2.83 × 10^7^ cells) compared to P40 ceramics (2.74 × 10^7^ cells). This suggests that the larger pore size of P45 allows for a higher yeast immobilization density. However, the weaker attachment of yeast cells to P45 results in easier detachment during washing. In contrast, the smaller pore size of P40 provides a more stable environment for yeast adhesion and retention, albeit with a slightly lower overall immobilization density.

To evaluate the potential of ceramics as fixatives to support yeast growth, yeast immobilized on each ceramic type (P40, P45, and the non-porous plate) was incubated in SD medium. As shown in Figure 3, the number of yeast cells immobilized on the porous ceramics (P40 and P45) increased significantly after 24 h of incubation. For P40, the initial immobilized yeast cell count was 0.15 × 10^7^ cells, which increased to 1.65 × 10^7^ cells after 24 h. Similarly, for P45, the yeast cell count increased from 0.17 × 10^7^ cells at 1 h to 1.47 × 10^7^ cells after 24 h. These results indicate that the porous structures of P40 and P45 ceramics provide a favorable environment for yeast proliferation, enhancing both adhesion and growth over time. Conversely, the non-porous ceramic plate exhibited minimal yeast growth, highlighting its limited capacity to support yeast immobilization and proliferation.

These findings suggest that a 24-h immobilization period supports robust yeast proliferation and retention in porous materials. While additional time points were not tested in this study, the marked increase in immobilized cell numbers between 1 and 24 h indicates that extended immobilization enhances colonization. Ceramics with P45, characterized by a larger pore size (300–1000 µm), enable better yeast cell attachment and subsequent growth compared to P40. However, the superior initial adhesion and retention of yeast cells by P40 ceramics make it more stable during washing. In contrast, non-porous ceramic plates lack these advantages, resulting in significantly lower yeast immobilization and growth. These insights informed the selection of PS as the next material for further studies, leveraging its high porosity and capacity to support robust yeast immobilization and growth.

#### 3.1.2. Yeast Immobilization on Polyurethane Sponges

The pore size of PS was measured as 500–1000 µm (Figure 4A). Dut to pore size and its high cell retention capacity, the PS was selected in this study. As shown in Figure 4A (left), the PS consists of a polyurethane sponge housed within a support structure, allowing for stability during the immobilization process. In contrast to the ceramic materials in Figure 1, the PS in Figure 4A (right) exhibits a highly interconnected, three-dimensional porous structure, which enhances fluid retention and provides a favorable environment for microbial immobilization.

The results of yeast cell distribution, including the number of cells in the washing solution, immobilized solution, and total immobilized yeast cells, are detailed in Figure 4B. The total number of yeast cells immobilized in the PS was 2.22 × 10^7^ cells, comprising 1.76 × 10^7^ immobilized cells and 0.46 × 10^7^ cells detected in the washing solution. In comparison, P45 exhibited a significantly lower total yeast cell count of 0.49 × 10^7^ cells, including 0.17 × 10^7^ immobilized cells and 0.32 × 10^7^ cells in the washing solution. In comparison, P45 exhibited a significantly lower total yeast cell count of 0.49 × 10^7^ cells, including 0.17 × 10^7^ immobilized cells and 0.32 × 10^7^ cells in the washing solution. This marked difference highlights the superior performance of the PS in creating an environment conducive to enhance yeast immobilization. The porous structure of PS likely enhances yeast immobilization by providing a larger surface area and protecting cells from detachment, leading to significantly higher retention compared to ceramics.

To assess the stability of the immobilized yeast cells, the PS remained at 1.70 × 10^6^ cells after 24 h in PBS, while the surrounding PBS solution had only 1.00 × 10^5^ cells, confirming strong retention. These results demonstrate the PS’s strong ability to retain immobilized yeast cells within its matrix. This high retention rate underscores the effectiveness of the PS as a yeast immobilization medium, maintaining a substantial portion of the yeast population within its porous structure. The exceptional retention capacity of the PS can be attributed to its high surface area, which facilitates yeast adhesion and proliferation, as well as its matrix structure, which likely provides a microenvironment that protects cells from shear forces during washing. These features contribute to reduced cell loss and enhance overall retention.

However, the specific mechanisms enabling the PS to achieve superior yeast immobilization and retention remain under investigation. This unique combination of properties led us to designate this system as a “yeast porous material complex (PS-complex)” reflecting its dual functionality and potential for further optimization.

### 3.2. Mo Adsorption on Yeast Porous Materials Complex

The Mo adsorption capacity of the PS-complex was evaluated and compared to that of free cells to assess its efficiency and robustness. The results revealed that, after 24 h, the PS-complex achieved an impressive 380.1 µg of Mo adsorption, comparable to the 338.4 µg adsorbed by ScBp6-free cells, while the PS alone exhibited negligible adsorption (Figure 5). Statistical analysis using one-way ANOVA confirmed that the differences in Mo adsorption between the PS-complex, free ScBp6 cells, and PS-only control were statistically significant (*p* < 0.05). This finding confirms that immobilizing yeast cells within the PS matrix offers a stable and practical alternative to free-cell systems by enabling easy handling, reusability, and potential scalability. The ability of the PS-complex to maintain adsorption efficiency despite immobilization is particularly noteworthy, as it suggests that the yeast cells remain functionally active within the porous structure. Since Mo is adsorbed in the form of MoO_4_^2−^, the adsorption capacity of 2.16 pg Mo per yeast cell corresponds to approximately 2.25 × 10^−14^ moles of Mo per yeast cell, equating to around 1.35 × 10^10^ Mo atoms per cell. Mo was introduced as sodium molybdate dihydrate (Na_2_MoO_4_·2H_2_O), which dissociates to release MoO_4_^2−^ ions under neutral pH. While ICP-MS quantifies total Mo, MoO_4_^2−^ is the dominant species in solution at the tested conditions. Therefore, adsorption is presumed to occur primarily in this form.

The relatively high adsorption of MoO_4_^2−^ suggests that the cell surface engineered yeast cell in PS-complex possesses highly accessible and effective binding sites for MoO_4_^2−^. The rapid equilibrium observed within 1 h further supports the notion that the binding sites are highly accessible and interact efficiently with MoO_4_^2−^ ions. Furthermore, the PS-complex exhibited rapid adsorption behavior, reaching equilibrium within the first hour, with adsorption levels ranging from 40.1% to 42.23%. This efficiency upon initial exposure to Mo highlights its potential for real-world applications requiring rapid rare metal recovery. The negligible adsorption observed in the PS-only control confirms that Mo recovery is driven by immobilized yeast rather than the sponge itself.

To further evaluate adsorption kinetics, Mo uptake by both the PS-complex and free cells was analyzed. Both systems demonstrated rapid adsorption, attaining equilibrium within 1 h. The PS-complex adsorbed 2.05 pg of Mo per cell at 1 h, with a slight increase to 2.16 pg per cell at 24 h, while free cells reached 2.15 pg per cell at 1 h, followed by a minor decrease to 1.96 pg per cell at 24 h (Figure 6). These results suggest that Mo adsorption primarily occurs in the initial phase and remains stable over time, with no significant difference between immobilized and free cells. The minimal adsorption beyond the first hour reinforces the conclusion that equilibrium is reached rapidly. Additionally, the negligible adsorption in the PS control further supports that Mo uptake is facilitated by yeast cells rather than the sponge matrix.

These findings underscore the outstanding potential of the PS-complex as a dual- function system, combining the advantages of yeast cell immobilization with high Mo adsorption capacity. The ability to maintain such performance highlights its suitability for scalable applications in resource recovery. The PS-complex offers a promising alternative to traditional free-cell approaches. Its stable and scalable nature makes it highly suitable for practical applications in resource recovery, providing an efficient platform for sustainable metal biosorption.

## 4. Discussion

### 4.1. Scaling up for Enhanced Mo Adsorption

The polyurethane sponge (PS)-complex used in this study was tested as a single unit cube, demonstrating a maximum Mo adsorption capacity of 40.1% within the first hour, stabilizing at 42.23% over 24 h. As demonstrated in our previous study [11], the wild-type strain BY4741 exhibited minimal Mo adsorption, and was therefore excluded from the present experiments focused on comparing the performance of engineered ScBp6 in free versus immobilized form. Each PS cube adsorbed approximately 380.1 µg of Mo, indicating that adsorption capacity is inherently limited by the available surface area for yeast immobilization and Mo binding [11,26,27,28]. To overcome this limitation and enhance total Mo recovery, scaling up the system by increasing the number of PS cubes is a logical next step. Since preliminary results indicate that Mo adsorption per yeast cell remains consistent when scaling from one to multiple PS cubes, it can be expected that increasing the number of PS cubes will proportionally increase overall Mo adsorption while maintaining efficiency. For example, if a single PS cube adsorbs 380 µg of Mo in 1 h, then deploying 10 PS cubes in parallel under similar conditions could theoretically adsorb 3.8 mg of Mo in the same time frame. This suggests that a modular approach, where multiple PS units are integrated into an adsorption system, could significantly enhance metal recovery capacity without compromising adsorption efficiency. Our previous study used calcium alginate as an immobilization matrix for ScBp6; however, it showed limited mechanical stability and lower reusability [11]. In contrast, PS offers structural resilience and enhanced cell retention, making it more suitable for long-term applications and continuous industrial-scale operations.

The results further show that MoO_4_^2−^ is adsorbed at approximately 2.25 × 10^−14^ moles per yeast cell, which is significantly higher than typical intracellular Mo levels in biological systems, often ranging between 10^−17^ and 10^−15^ moles per cell. The Mo accumulation capacity of *Rhodococcus* strains, estimated from experimental conditions reported by Ivshina et al. [29], suggests that each bacterial cell could accumulate approximately 6.67 × 10^−15^ moles of Mo per cell. This comparison highlights that the engineered yeast in this study exhibits more than three times the Mo adsorption capacity per cell compared to *Rhodococcus*. Moreover, unlike *Rhodococcus*, which partially converts Mo into intracellularly stored or nanoparticle-precipitated forms, the yeast is presumed to retains MoO_4_^2−^ on its surface, based on the rapid adsorption kinetics and prior findings with cell-surface display systems. However, further investigation using surface-specific analytical techniques is warranted to confirm the exact localization. The strong binding of MoO_4_^2−^ to the engineered yeast surface enhances the stability of the PS-complex, preventing desorption and ensuring efficient Mo recovery. These findings further support the advantage of yeast-based immobilization strategies for scalable and practical Mo bio-recovery applications. Compared to conventional Mo recovery methods, such as ion exchange, solvent extraction, and chemical precipitation, which often involve high energy consumption or generate secondary waste, the yeast-based biosorption approach offers a sustainable, reusable, and cost-effective alternative with competitive adsorption capacity.

### 4.2. Full Scale System for Practical Application

To translate the laboratory-scale findings of the PS-complex into a practical, full-scale system, a promising approach is to integrate the PS-complex into a DHS system, designed for continuous wastewater treatment and Mo recovery. The use of PS as a support medium has proven advantages in wastewater treatment, as seen in the DHS reactor, where it effectively promotes biomass proliferation for pollutant removal [25,30]. In this configuration, the PS-complexes would be suspended within a vertical column, with wastewater flowing down through the column and coming into contact with the PS material, allowing Mo ions to adsorb to the yeast-immobilized sponges. The downflow design ensures uniform distribution of wastewater across the PS-complexes, maximizing the surface area exposed to contaminants and facilitating efficient Mo adsorption. This continuous treatment process offers advantages for large-scale applications, enabling a steady recovery of Mo from wastewater. By adjusting the number of PS units or columns in the system, the Mo adsorption capacity can be scaled up according to specific requirements, whether dealing with varying wastewater volumes or Mo concentrations.

Furthermore, the modular nature of the DHS system enhances its flexibility, allowing for easy expansion or modification to meet changing demands. The DHS process itself has evolved as an energy-efficient alternative to the activated sludge process, which is widely used but has limitations, such as high-energy consumption and large amounts of excess sludge [25]. The transition of the DHS process from research to real-world deployment is already underway in countries such as India, Egypt, and Japan, highlighting its potential for broad application in wastewater treatment. Additionally, biogenic manganese oxides (bio-MnO_2_) have been shown to absorb minor metals, and bioreactor cultivation of manganese-oxidizing bacteria (MnOB) offers potential for removing and recovering minor metals from wastewater. The integration of immobilized systems for metal recovery has also been widely recognized in bacterial bioremediation platforms, where physical confinement improves cell stability, reusability, and resistance to harsh wastewater environments [22]. These findings further support the practicality of applying immobilized microbial systems. like the PS-complex. in continuous treatment setups. In a DHS reactor, the simultaneous removal of metals like Ni and Co has been achieved, further supporting the viability of using such systems for metal recovery [31]. Similarly, a novel technology combining amino organic ligands with sewage sludge has demonstrated effective removal of heavy metals, like Cu, Cr, and Zn, as well as fecal coliforms, showing the potential of modified bioactive sorbent systems for enhancing wastewater treatment and pollutant recovery [32]. These findings support the idea that integrating PS-complexes into a DHS system could offer broad and effective solutions for pollutant removal in various wastewater applications. However, for successful full-scale implementation, several critical factors must be addressed. First, hydraulic flow and pressure drop within the system must be optimized to ensure even wastewater distribution across all PS units without excessive resistance, which could negatively impact adsorption efficiency. Additionally, the long-term durability and stability of the PS-complexes must be evaluated, as continuous exposure to wastewater may lead to degradation or clogging of the porous material. Strategies for regenerating or replacing saturated PS-complexes will also be essential to maintaining system performance and reducing operational costs.

Moreover, the design of the DHS system should incorporate monitoring and control mechanisms to track Mo adsorption rates, ensure uniform wastewater distribution, and detect potential issues, such as clogging or material wear. This will allow for optimization of adsorption rates and ensure long-term system efficiency. Despite these challenges, integrating PS-complexes into a DHS system holds great promise for large-scale, sustainable Mo recovery. Future research should focus on optimizing the design and operational parameters of the DHS system, as well as evaluating its performance under practical conditions.

## 5. Conclusions

This study evaluated the immobilization and Mo adsorption capabilities of yeast cells on porous ceramics (P40 and P45) and PS, providing insights into the influence of material structure on performance. Porous ceramics supported effective yeast immobilization compared to non-porous plates. P45 (300–1000 µm pores) showed higher immobilization densities. Both P40 and P45 significantly enhanced yeast proliferation, demonstrating the benefits of porous structures for microbial growth. However, PS outperformed ceramics in yeast immobilization and Mo adsorption. The PS retained 2.22 × 10^7^ yeast cells, including 1.76 × 10^7^ immobilized cells and 0.46 × 10^7^ cells in the washing solution. It achieved rapid Mo adsorption, stabilizing at 40.1% within the first hour, with an Mo adsorption capacity of 2.16 pg/yeast cell, comparable to free ScBp6 cells (1.92 pg/yeast cell). These findings underscore the importance of material properties, such as pore size and structure, in optimizing yeast immobilization, retention, and adsorption performance. The PS complex, shown in Figure 7, demonstrated a highly porous structure and superior yeast cell retention, positioning it as an exceptional biosorption platform for resource recovery. Its scalability for enhanced Mo recovery and potential integration into full-scale systems, such as the DHS system, further underscore its practical applicability. Future research should investigate the underlying mechanisms governing yeast retention in PS and explore material modifications to further optimize Mo recovery. In addition, future studies should explore fine-tuning of immobilization conditions (e.g., cell density, incubation time), investigate the influence of co-existing ions and organic matter in real wastewater, and develop advanced surface-display systems, with enhanced binding affinity through targeted genetic modifications. In summary, PS offers significant advantages for scalable, sustainable yeast immobilization and Mo recovery, highlighting its promise for resource recovery applications.

## Figures and Tables

**Figure 1 microorganisms-13-01034-f001:**
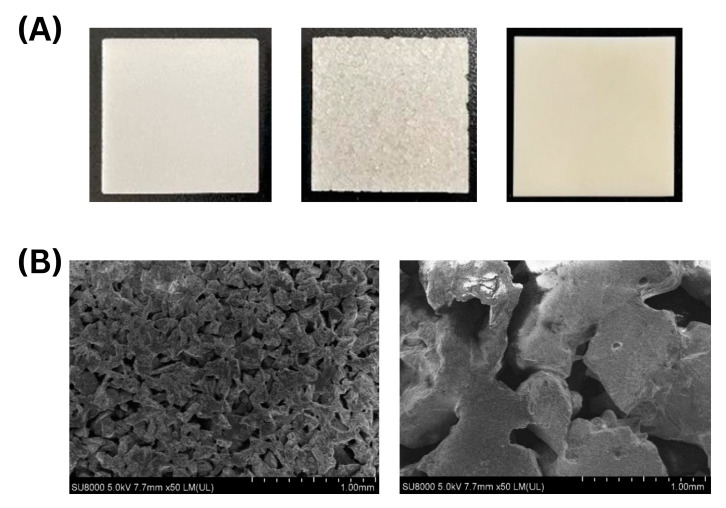
Macroscopic and microscopic characterization of porous materials (P40, P45, and Plate). (**A**) Digital photographs of ceramic materials P40 (**left**), P45 (**middle**), Plate (**right**) (25 mm × 25 mm). (**B**) SEM images of the porous materials at 5.0 kV, 7.7 mm × 50 LM., showing surface morphologies of P40 (**left**) and P45 (**right**). Scale bars: 1 mm.

**Figure 2 microorganisms-13-01034-f002:**
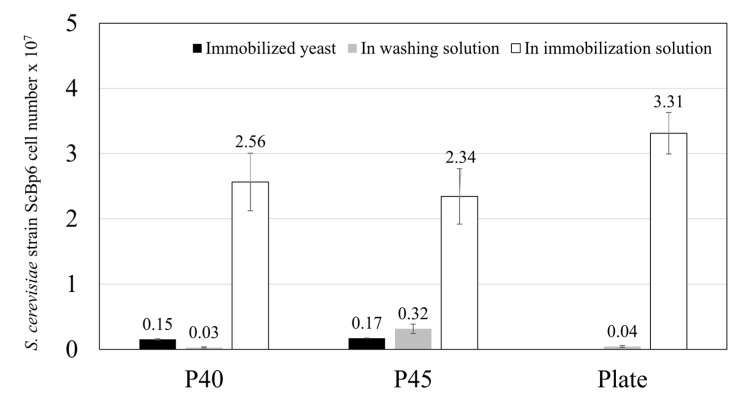
Comparison of ScBp6 immobilization on ceramic materials with different pore sizes (P40 and P45). Yeast adhesion and proliferation were evaluated by counting the number of immobilized cells after 24 h.

**Figure 3 microorganisms-13-01034-f003:**
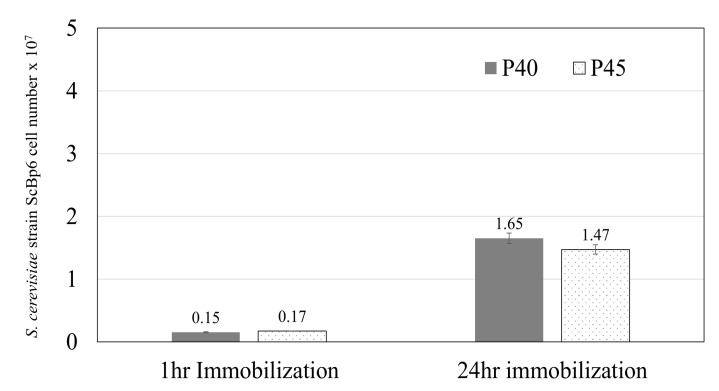
Proliferation of ScBp6 cells immobilized on P40 and P45 ceramics after 24 h of incubation. Both porous ceramics supported yeast growth, while non-porous ceramic plates showed minimal proliferation.

**Figure 4 microorganisms-13-01034-f004:**
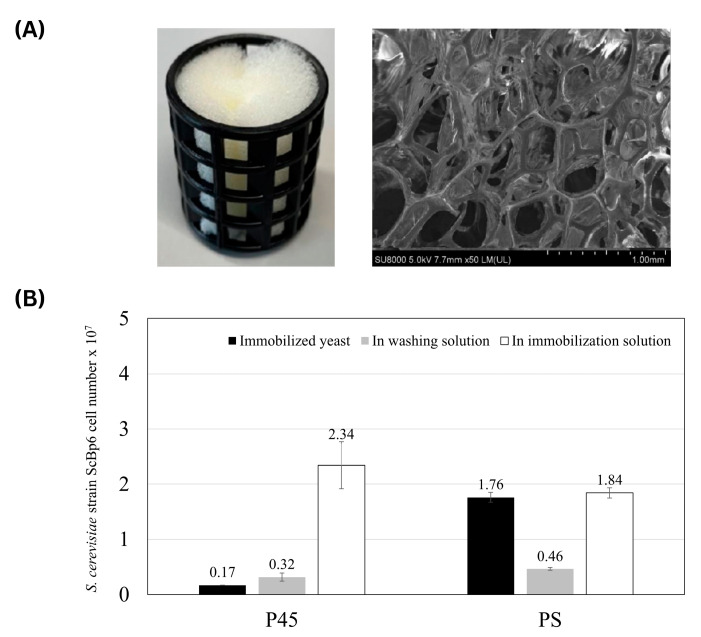
Immobilization of yeast cells in PS. (**A**) Images of the PS used in this study. The (**left**) panel shows the PS housed within a support structure while the (**right**) panel presents a SEM image of the PS structure (**B**) Immobilization efficiency of ScBp6 on PS shows that the number of immobilized cells in PS was significantly higher than in ceramic materials, indicating superior yeast retention.

**Figure 5 microorganisms-13-01034-f005:**
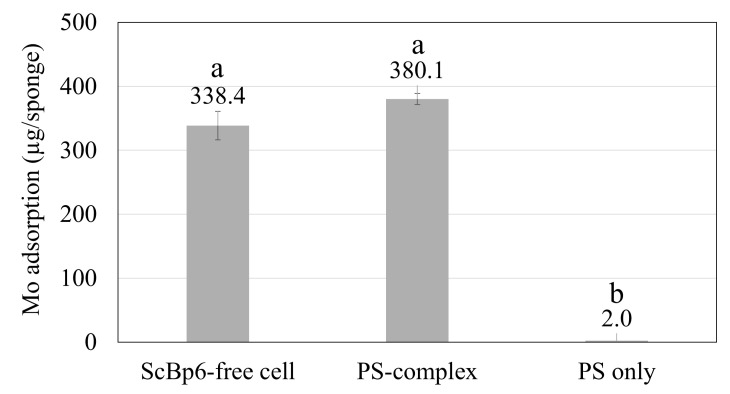
Mo adsorption (µg per sponge) for three conditions: free ScBp6 cells (equivalent cell number) yeast-immobilized PS (PS-complex), and PS without cells (control). Adsorption was measured after 24 h. Data represent mean ± SD (*n* = 3). Different letters indicate significant difference where *p* < 0.05. Bars which have no common letters are significantly different.

**Figure 6 microorganisms-13-01034-f006:**
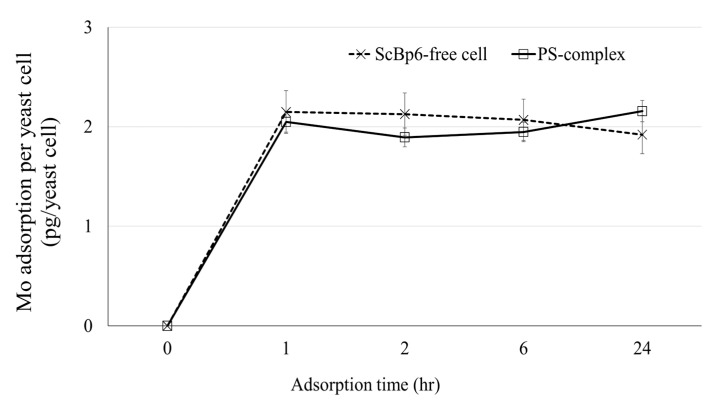
Mo adsorption kinetics per yeast cell (pg Mo/yeast cell) for PS-complex (ScBp6-immobilized) and ScBp6-free cells. Adsorption was monitored over 24 h in 10 ppm Mo solution. PS-only control showed negligible adsorption and is omitted for clarity. Data represent mean ± SD (*n* = 3).

**Figure 7 microorganisms-13-01034-f007:**
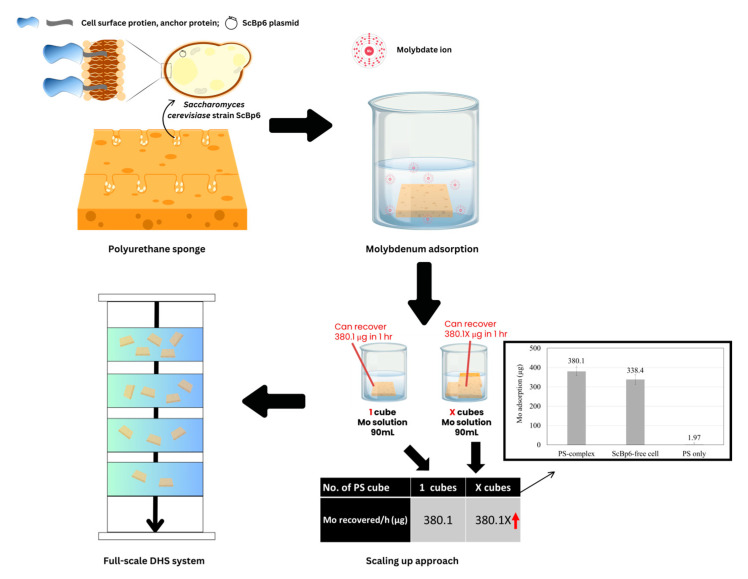
Schematic representation of PS-complex yeast immobilization and Mo adsorption for scalable recovery systems.

## Data Availability

The raw data supporting the conclusions of this article will be made available by the authors on request.

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
