# Peer review of "Enhanced Molybdenum Recovery Achieved by a Complex of Porous Material-Immobilized Surface-Engineered Yeast in Development of a Sustainable Biosorption Technology"

_microorganisms, 2025, doi:10.3390/microorganisms13051034_

Round 1
Reviewer 1 Report
Comments and Suggestions for Authors
Dear Authors,
The manuscript explores the biosorption of molybdenum using a surface-engineered yeast (Saccharomyces cerevisiae strain ScBp6) immobilized in porous materials. The study is relevant given the growing interest in sustainable recovery methods for critical metals such as Mo. The experiments are well-designed, and the results support the conclusions of the paper. However, there are several areas that could be improved for greater clarity and rigor.
Majors:
- The way references are cited in the text does not follow this journal's format. Please change them.
- In the introduction, it is not clear why Saccharomyces cerevisiae is being used, as far as I know, this organism does not metabolize molybdenum. So, why would it bioabsorb it? I don't understand. Please explain this in the introduction.
-In the introduction, it would be helpful to include more general information about the engineered yeast strain (ScBp6) and its mechanism of interaction with Mo
- L209: "A 24-hour immobilization period is optimal." Why not select another intermediate time point? I believe that claiming it is "optimal" based on just two time points is insufficient.
- In the figure captions, the number of times each experiment was repeated should be included, along with the statistical error, to determine whether the differences are statistically significant.
-Justify the Mo concentrations used in the adsorption experiments
-L241: “To assess the stability…..” Sorry but, where are those data??
-L270:” molybdate (MoO2−)???” molybdate (MoO42−)
- In Figure 5, it is essential to determine whether those differences are statistically significant.
- In my opinion, the units in Figure 5 should be the percentage of molybdenum immobilized rather than micrograms. Please change them and discuss accordingly
- I'm afraid that Figure 6 cannot be used as it is to derive anything significant from it; it doesn't even include error bars.
L294: “The PS-complex offers a promising alternative to traditional free-cell approaches” In a previous study from 2020, the authors conducted a similar experiment but immobilized the yeasts in a calcium alginate matrix. In my opinion, they should have included this material as a reference in these experiments as well. Why didn’t they do so? Please discuss.
-L333: “DHS system” please define what is this.
-In the discussion, elaborate further on the implications of the findings for real-world applications in industrial wastewater treatment.
-Compare the performance of the biosorption method developed in this study with other Mo recovery technologies.
-In the discussion, highlight areas for future research, such as the optimization of immobilization conditions, the assessment of the impact of various contaminants on Mo biosorption, and the exploration of genetic engineering potential to further enhance the yeast's Mo-binding capabilities
Minors:
-L147: “.?? Yeast immobilization” typo
Author Response
Reviewer 1
Dear Reviewer,
We sincerely thank you for the thoughtful and constructive feedback provided on our manuscript entitled “Enhanced Molybdenum Recovery Achieved by Surface-Engineered Yeast Immobilized on Porous Material Complex: Development of a Sustainable Biosorption Technology.”
We deeply appreciate the insightful comments and suggestions, which have significantly improved the clarity, structure, and scientific rigor of our work. We have carefully addressed each point raised and revised the manuscript accordingly. Below, we provide detailed responses to each comment. For transparency, we have also indicated the specific line numbers in the revised manuscript where changes have been made.
We hope that the revised version meets your expectations and look forward to your further evaluation.
Best regards,
Comment 1: The way references are cited in the text does not follow this journal's format. Please change them.
Response: Thank you for pointing this out and we apologize for our careless mistake. We have updated the reference formatting accordingly as required by Microorganisms. Line 546-616
Comment 2: In the introduction, it is not clear why Saccharomyces cerevisiae is being used, as far as I know, this organism does not metabolize molybdenum. So, why would it bio-absorb it? I don't understand. Please explain this in the introduction.
Response: Thank you for your valuable comment. We have clarified this point by highlighting that S. cerevisiae serves as an ideal platform for surface protein engineering due to its robustness, genetic tractability, and safety status. “Saccharomyces cerevisiae provides an advantageous platform for surface engineering due to its genetic tractability and robustness under diverse environmental conditions, facilitating practical biosorption applications. Besides, S. cerevisiae doesn’t metabolize Mo, so the effect of engineered yeast on Mo biosorption is clear.” from Line 76-81.
Comment 3: In the introduction, it would be helpful to include more general information about the engineered yeast strain (ScBp6) and its mechanism of interaction with Mo.
Response: We appreciate the reviewer’s suggestion. In the revised manuscript, we have expanded the Introduction to include a clearer explanation of the engineered yeast strain ScBp6 and the function of the ModE protein. Specifically, we now describe that ScBp6 displays the molybdate-binding domain of the ModE regulator on its surface, which enables selective adsorption of MoO₄²⁻ ions. This addition clarifies the underlying mechanism of Mo biosorption by the engineered yeast. Line 111-113.
Comment 4: L209: "A 24-hour immobilization period is optimal." Why not select another intermediate time point? I believe that claiming it is "optimal" based on just two time points is insufficient.
Response: Thank you for pointing this out. Though we have investigated several time period and chosen the 24-hour immobilization period as the best one, we agree that "optimal" may be too strong given the limited time points tested. We have revised the statement in Section 3.1.1 to soften the language. The revised text now emphasizes that 24 hours promoted robust proliferation, while acknowledging that additional time points would be needed to confirm optimality. Line 290-293.
Comment 5: In the figure captions, the number of times each experiment was repeated should be included, along with the statistical error, to determine whether the differences are statistically significant.
Response: We thank the reviewer for this insightful suggestion. In the revised manuscript, we have added a new subsection titled “2.5. Statistical Analysis” under the Materials and Methods section to clarify our statistical approach. This section states that all experiments were conducted in triplicate (n = 3), and results are reported as mean ± standard deviation (SD). One-way ANOVA was used to assess statistical significance (p < 0.05). Line 221-224.
Comment 6: Justify the Mo concentrations used in the adsorption experiments
Response: We thank the reviewer for this suggestion. We have now clearly explained that the selected 10 ppm Mo concentration reflects upper-range concentrations commonly found in industrial wastewater: "The concentration of 10 ppm Mo was chosen to simulate upper-range industrial wastewater concentrations (typically reported between 10 µg/L and 10 mg/L), as previously documented [5,7].” Line 205-207
Comment 7: L241: “To assess the stability…..” Sorry but, where are those data??
Response: Thank you for pointing out this oversight. We have added explicit quantitative data confirming yeast stability in the PS matrix: “the PS remained at 1.70 × 106 cells after 24 hours in PBS, while the surrounding PBS solution had only 1.00 × 105 cells, confirming strong retention.” Line 325-327
Comment 8: L270:” molybdate (MoO2−)???” molybdate (MoO42−)
Response: Thank you for catching this typographical error. We have corrected it throughout the manuscript, such as Line 356, 359, 360, 362, 363, and 365
Comment 9: In Figure 5, it is essential to determine whether those differences are statistically significant.
Response: We appreciate this observation. To address this comment, we have now explicitly stated in the Results section (Section 3.2) that the differences in Mo adsorption values presented in Figure 5 were statistically significant. This clarification complements the previously added details in the Statistical Analysis section and provides clearer support for the trends observed in Figure 5. Line 349-351.
Comment 10: In my opinion, the units in Figure 5 should be the percentage of molybdenum immobilized rather than micrograms. Please change them and discuss accordingly.
Response: We thank the reviewer for this insightful suggestion. While we initially considered changing the y-axis of Figure 5 to percentage, we chose instead to modify the unit to µg/sponge to highlight the absolute Mo adsorption capacity per polyurethane sponge unit. This presentation enables clearer discussion on the scalability of the system for future practical applications.
To address the reviewer’s intent, we have included percentage-based adsorption efficiency values (40.1–42.23%) in the main text (Section 3.2) for comparative clarity. We believe that maintaining the current unit allows us to convey both the system’s performance and scale-up potential effectively. Line 367 and Figure 5
Comment 11: I'm afraid that Figure 6 cannot be used as it is to derive anything significant from it; it doesn't even include error bars.
Response: Thank you for highlighting this important point. In response, we have revised Figure 6 to include error bars representing standard deviation across triplicate experiments (n = 3). This ensures a clearer interpretation of variability and supports the reliability of the adsorption kinetics data. Line 395
Comment 12: L294: “The PS-complex offers a promising alternative to traditional free-cell approaches” In a previous study from 2020, the authors conducted a similar experiment but immobilized the yeasts in a calcium alginate matrix. In my opinion, they should have included this material as a reference in these experiments as well. Why didn’t they do so? Please discuss.
Response: We appreciate your suggestion. We have now explicitly mentioned our previous calcium alginate-based immobilization study, highlighting why polyurethane sponge (PS) presents advantages. We addressed "Our previous study used calcium alginate as an immobilization matrix for ScBp6 [11], however, it showed limited mechanical stability and lower reusability. In contrast, PS offers structural resilience and enhanced cell retention, making it more suitable for long-term applications and continuous industrial-scale operations." Line 414-418
Comment 13: L333: “DHS system” please define what is this.
Response: Thank you for this comment. We have clearly defined the DHS system upon its first mention: “Downflow Hanging Sponge (DHS), a biofilm reactor widely applied for wastewater treatment.” Line 179-181
Comment 14: In the discussion, elaborate further on the implications of the findings for real-world applications in industrial wastewater treatment.
Response: We have expanded Section 4.2 of the Discussion to highlight the real-world applicability of the PS-complex for Mo recovery. We describe how this system could be integrated into a Downflow Hanging Sponge (DHS) reactor, a biofilm-based system already applied in wastewater treatment in countries like Japan and India. We also outline design and operational considerations for scale-up, such as PS durability, hydraulic flow optimization, and system monitoring. Line 472-481
Comment 15: Compare the performance of the biosorption method developed in this study with other Mo recovery technologies.
Response: We thank the reviewer for this important suggestion. We have expanded the discussion in Sections 4.1 to highlight comparisons between our biosorption method and other molybdenum recovery technologies. Specifically, we emphasize the superior adsorption per cell compared to Rhodococcus strains, and we discuss the broader advantages of yeast-based immobilization over conventional methods such as ion exchange and precipitation. These revisions demonstrate the practical and technical merits of our system. Line 421-426
Comment 16: In the discussion, highlight areas for future research, such as the optimization of immobilization conditions, the assessment of the impact of various contaminants on Mo biosorption, and the exploration of genetic engineering potential to further enhance the yeast's Mo-binding capabilities
Response: We appreciate the reviewer’s recommendation. In the revised Conclusion section, we have expanded the future research outlook to specifically mention the need for optimizing immobilization conditions, evaluating the influence of wastewater co-contaminants, and exploring further genetic engineering to enhance Mo-binding affinity. These directions will help guide the next phase of development for practical implementation. Line 508-512.
Comment 17: L147: “.?? Yeast immobilization” typo
Response: Thank you for pointing out our mistyping, which have been corrected already. Thank you for noting this. Line 226
Reviewer 2 Report
Comments and Suggestions for Authors
The manuscript entitled “Enhanced Molybdenum Recovery Achieved by Surface-Engineered Yeast Immobilized on Porous Material Complex: Development of a Sustainable Biosorption Technology” is devoted to Mo biosorption using yeast Saccharomyces cerevisiae strain ScBp6 immobilized on porous material for its recovery from industrial wastewater.To my mind this manuscript is corresponding to the aims and scopes of the “Microorganisms” journal. I am ready to recommend it for publication after some corrections, due to the comments below.
- In the abstract, it is necessary to clarify what surface-engineered yeast is
- It is not clear how work 18 relates to the article, it should either be removed or explained in more detail
- In the introduction, more attention should be paid to porous materials and their role in yeast immobilization
- In the introduction, it is necessary to dwell in more detail on surface-engineered yeast and their use
- In the objective, it is necessary to specify porous materials
- It is necessary to describe the experimental design
- It is necessary to describe the equipment used in the work, including the manufacturer and its operating mode
- Since only 2 materials were used in the work, it is worth explaining their choice. Has a screening of different materials for cell adhesion been carried out before?
- The data in Figure 1 are uninformative, especially the first part
- It is not very clear from the methodology how Yeast adhesion and proliferation were obtained
- Since Mo is adsorbed in the form 269 of molybdate (MoO2−), how is this proven?
- If the authors claim that engineered yeast cells extract more molybdenum, it would be worth conducting similar experiments with a conventional strain
- Figures 5 and 6 show the kinetics for which material? It is worth providing data for both materials with and without cells. If I understand correctly, loading the material with cells does not greatly affect molybdenum immobilization. Or the figures need to have clearer captions.
- 327 MoO42− on its surface no evidence
- If the authors describe an approach to scaling their data in the form of a setup, its description and diagram are required. It is advisable to provide economic calculations for using the setup to isolate molybdenum from wastewater. In addition, it is not clear whether the authors used a model of wastewater or real water.
- Figure 7 is more suitable for a graphic abstract.
Author Response
Reviewer 2
Dear Reviewer,
We sincerely thank you for the thoughtful and constructive feedback provided on our manuscript entitled “Enhanced Molybdenum Recovery Achieved by Surface-Engineered Yeast Immobilized on Porous Material Complex: Development of a Sustainable Biosorption Technology.”
We are grateful for your careful reading and helpful suggestions, which have allowed us to significantly improve the clarity, experimental detail, and real-world relevance of the manuscript. We have carefully revised the text and figures in accordance with your comments and have provided point-by-point responses below. Line numbers corresponding to each revision are also included for clarity.
We appreciate your recommendation for publication and hope the revised version meets your expectations.
Best regards,
Comment 1: In the abstract, it is necessary to clarify what surface-engineered yeast is
Response: We appreciate the reviewer’s comment. In the revised abstract, we have clarified the term "surface-engineered yeast" and specified the strain ScBp6 displays a molybdate-binding protein (ModE) on its cell surface. This addition improves the clarity of our approach from the outset. Line 21-24
Comment 2: It is not clear how work 18 relates to the article, it should either be removed or explained in more detail
Response: Thank you for your suggestion. After revising, we have removed the reference [18] and replaced with the clearer reference related to the challenges of stability and performance reduction. Line 92-95
Comment 3: In the introduction, more attention should be paid to porous materials and their role in yeast immobilization
Response: Thank you for your helpful suggestion. In the revised introduction, we have added a description of the role of porous materials in microbial immobilization. We now explain how pore structure influences yeast adhesion, proliferation, and retention, which are critical for the biosorption system’s performance. Line 95-106
Comment 4: In the introduction, it is necessary to dwell in more detail on surface-engineered yeast and their use
Response: We thank the reviewer for this comment. In response, we have expanded the Introduction to provide a clearer explanation of surface-engineered yeast. We now describe how the ScBp6 strain was constructed by displaying the ModE molybdate-binding protein on the cell surface, enabling targeted biosorption of Mo. This addition clarifies both the mechanism and the rationale behind using engineered yeast in our system. Line 61-73
Comment 5: In the objective, it is necessary to specify porous materials
Response: Thank you for this helpful suggestion. We have revised the objective statement in the Introduction to explicitly state the types of porous materials used in this study which are ceramic plates and polyurethane sponge. This clarification helps readers better understand the experimental scope and rationale from the outset. Line 108-111
Comment 6: It is necessary to describe the experimental design
Response: Thank you for this helpful suggestion. We have added a new subsection (2.2. Experimental Design) to clearly describe the overall experimental design through this study. This paragraph outlines the sequential steps of material preparation, yeast immobilization, surface characterization, Mo adsorption testing, and statistical analysis, thereby helping readers understand the flow and scope of the experiments. Line 138-153
Comment 7: It is necessary to describe the equipment used in the work, including the manufacturer and its operating mode
Response: We appreciate the reviewer’s suggestion. We have revised the Materials and Methods section to include the specific models and manufacturers of the equipment used, including;
- WST assay kit (Dojindo, M439),
- Varioskan™ LUX plate reader (Thermo Fisher Scientific, USA)..
- ICP-MS Nexion, (Perkin Elmer, Waltham, MA, USA)
- SEM (SU8000, Hitachi High-Tech Co., Tokyo, Japan) which is operated at 5.0 kV, 7.7 mm × 50 LM
These updates improve clarity and reproducibility. Line 134-136 and Line 149-152 and Line 260
Comment 8: Since only 2 materials were used in the work, it is worth explaining their choice. Has a screening of different materials for cell adhesion been carried out before?
Response: Thank you for your comment. We have added a brief explanation to clarify that ceramic materials were initially screened due to their structural variation, but demonstrated limited yeast retention. Polyurethane sponge (PS) was subsequently chosen based on its proven application in wastewater treatment bioreactors and favorable porous architecture. This rationale is now included in the Materials and Methods section to improve clarity regarding material selection. Line 139-146
Comment 9: The data in Figure 1 are uninformative, especially the first part
Response: Thank you for this comment. While we agree that pore structure details are better conveyed by SEM, we retained Figure 1A to illustrate the physical formats and size of the ceramic materials used. To clarify its role, we have updated the caption to emphasize that the digital photographs provide context for the experimental materials. Line 231-234
Comment 10: It is not very clear from the methodology how Yeast adhesion and proliferation were obtained
Response: Thank you for pointing this out. We have revised Sections 2.3.1 and 2.3.2 to clarify how yeast adhesion and proliferation were evaluated. Specifically, we explain that cell numbers were quantified using a standard curve based on WST absorbance and microscopic counts. Adhesion was assessed at 1 hour post-immobilization, and proliferation was determined by comparing immobilized cell counts before and after a 24-hour incubation period. Line 170-176 and Line 187-194
Comment 11: Since Mo is adsorbed in the form of molybdate (MoO₄²⁻), how is this proven?
Response: We thank the reviewer for this important question. In our experiments, Mo was introduced as sodium molybdate dihydrate (Na₂MoO₄·2H₂O), which dissociates into MoO₄²⁻ under neutral pH. After incubation with yeast, we centrifuged the mixture to separate the cells and collected the supernatant for ICP-MS analysis. A significant decrease in MoO₄²⁻ concentration in the supernatant was observed, indicating that MoO₄²⁻ were removed from solution, presumably due to adsorption by the yeast cells. Although this suggests that MoO₄²⁻ is the primary adsorbed species, we acknowledge that the ICP-MS method measures total Mo and does not distinguish speciation. Accordingly, we have softened the statement in the Results section and noted the need for surface-specific or speciation techniques in future work. Line 211-212 and Line 358-361
Comment 12: If the authors claim that engineered yeast cells extract more molybdenum, it would be worth conducting similar experiments with a conventional strain
Response: thank the reviewer for this suggestion. In our previous study (Stephanie et al., 2020, reference [11]), we evaluated molybdenum adsorption by wild-type S. cerevisiae strain BY4741 alongside engineered strains ScBp5 and ScBp6. The results demonstrated that the wild-type exhibited negligible adsorption capacity compared to engineered yeast. Therefore, in this study, we focused on evaluating the performance of ScBp6 in immobilized versus free-cell configurations. A clarifying sentence and citation have been added to the Discussion section to explain this design decision. Line 400-403 and Line 414-418
Comment 13: Figures 5 and 6 show the kinetics for which material? It is worth providing data for both materials with and without cells. If I understand correctly, loading the material with cells does not greatly affect molybdenum immobilization. Or the figures need to have clearer captions.
Response: Thank you for your comment. We have revised the captions for Figures 5 and 6 to clearly indicate which materials were tested, the corresponding sample conditions (with or without yeast), and the nature of the adsorption data. In particular, we clarified that PS-only controls were included and exhibited negligible adsorption, and this is now reflected in both the figure captions and Results section. These improvements help distinguish the contributions of immobilized yeast cells versus sponge matrix alone. Line 346-349 and Line 351-355
Comment 15: MoO42− on its surface no evidence
Response: We thank the reviewer for this comment. If we understand the question correctly, our engineered yeast expresses the ModE protein on the cell surface, and based on the biosorption mechanism, Mo is presumed to bind externally via the displayed binding domain. In support of this, our previous study (Stephanie et al., 2020 [11] and Chien et al., 2017 [21]) reported surface adsorption of Mo using a similar surface-engineered yeast platform. Additionally, in unpublished follow-up experiments, we digested the yeast surface proteins after Mo adsorption and detected MoO42− in the resulting supernatant, further suggesting surface retention. While these findings support our assumption, we acknowledge that direct surface-specific techniques such as SEM-was not applied in this study. We have revised the manuscript to soften the language accordingly and have indicated that direct surface analysis will be pursued in future work. Line 427–430
Comment 16: If the authors describe an approach to scaling their data in the form of a setup, its description and diagram are required. It is advisable to provide economic calculations for using the setup to isolate molybdenum from wastewater. In addition, it is not clear whether the authors used a model of wastewater or real water.
Response: Thank you for this valuable feedback. We have clarified that the adsorption experiments were conducted using a model solution prepared from sodium molybdate dihydrate in deionized water, simulating molybdenum-containing industrial wastewater. We also acknowledge that economic feasibility analysis and a reactor system schematic are beyond the scope of this initial study. However, we have added a statement in the Discussion indicating that these elements, including validation in real wastewater, pilot-scale testing, and cost analysis, are critical next steps for practical implementation. Line 472-481
Comment 17: Figure 7 is more suitable for a graphic abstract.
Response: We thank the reviewer for this suggestion. Figure 7 was included to visually summarize the proposed application and scale-up strategy. We agree that it could serve as an effective graphical abstract. However, based on our experience, graphical abstracts are often displayed only on the journal’s website and not included in the PDF version of the article. Because we believe this figure provides a useful overview of the research for all readers, we have retained it in the main manuscript. If appropriate, we believe that this figure may also serve as the graphical abstract for the article.
Round 2
Reviewer 1 Report
Comments and Suggestions for Authors
I believe the authors have adequately addressed all of my comments and suggestions, and I accept the paper in its current version.
Reviewer 2 Report
Comments and Suggestions for Authors
In my opinion, the authors have significantly revised the manuscript and taken into account all my comments. I am ready to recommend the manuscript for publication in this form.